# Ipragliflozin Ameliorates Endoplasmic Reticulum Stress and Apoptosis through Preventing Ectopic Lipid Deposition in Renal Tubules

**DOI:** 10.3390/ijms21010190

**Published:** 2019-12-26

**Authors:** Kohshiro Hosokawa, Tomoaki Takata, Takaaki Sugihara, Tomomitsu Matono, Masahiko Koda, Tsutomu Kanda, Sosuke Taniguchi, Ayami Ida, Yukari Mae, Marie Yamamoto, Takuji Iyama, Satoko Fukuda, Hajime Isomoto

**Affiliations:** 1Division of Medicine and Clinical Science, Faculty of Medicine, Tottori University, Yonago, Tottori 683-8504, Japan; 2Hino Hospital, Hino, Tottori 689-4504, Japan

**Keywords:** SGLT2 inhibitor, ER stress, lipotoxicity, steatonephropathy, ectopic fat accumulation, NASH, NAFLD

## Abstract

Background: Chronic kidney disease (CKD) and non-alcoholic steatohepatitis (NASH) are major health burdens closely related to metabolic syndrome. A link between CKD and NASH has been assumed; however, the underlying mechanism is still unknown. Ectopic lipid deposition (ELD) in the hepatocyte results in endoplasmic reticulum (ER) stress, which plays an important role in the development of steatohepatitis. ELD is also assumed to play a role in the development of kidney injury. We aimed to investigate the role of ELD and ER stress in the development of CKD, and evaluate the efficacy of a sodium glucose cotransporter-2 inhibitor, ipragliflozin. Methods: Male FLS-*ob/ob* mice that closely imitate the pathophysiology of NASH were treated with vehicle or ipragliflozin. Metabolic characteristics, histology of the kidney, ER stress, and apoptotic signals were evaluated. Results: The serum triglyceride was significantly lower in mice treated with ipragliflozin. Ipragliflozin reduced ELD in renal tubules. Ipragliflozin also reduced the expression levels of GRP78 and CHOP, apoptotic cells, and interstitial fibrosis. Conclusions: ELD induced kidney injury through ER stress. Ipragliflozin improved the pathogenesis of CKD by reducing ELD and ER stress in NASH-model mice. Our results suggest ipragliflozin has therapeutic effect on CKD in NASH.

## 1. Introduction

Chronic kidney disease (CKD) is one of the major public health issues in the world. CKD is a risk factor for cardiovascular disease and cerebrovascular diseases as well as end-stage renal diseases [1]. Metabolic syndromes—including diabetes, hypertension, obesity, and dyslipidemia—are strongly associated with the development and progression of CKD, and diabetic nephropathy is a well-known cause of end-stage renal diseases. Several peptidyl hormones mainly produced in the gastrointestinal tract are reportedly associated with the development of CKD in the context of diabetic nephropathy [2]. Glucagon-like peptide-1 and its related peptide, dipeptidyl peptidase IV, are the major therapeutic targets for diabetes and diabetic nephropathy [3]. In contrast with the current understanding and progression in therapeutic approach to diabetic nephropathy, mechanisms of CKD caused by dyslipidemia remain to be clarified. Recent studies have raised an issue concerning the association of dyslipidemia and the development of CKD [4,5]; however, the detailed mechanism linking lipid metabolism and kidney injury is not well known.

Non-alcoholic steatohepatitis (NASH) is the most common chronic liver disease, which develops without a history of excessive alcohol consumption [6]. A number of clinical investigations have reported the link between NASH and the progression of CKD [7,8]. Epidemiological evidence and the similarity in risk factors between NASH and CKD suggest the presence of common pathogenic mechanisms underlying both diseases. The pathological findings in NASH are hepatic steatosis with intralobular inflammation and ballooning degeneration of hepatocytes [9]. Ectopic lipid deposition (ELD) is toxic to hepatocytes and causes hepatocyte injury and inflammation [10]. ELD in the kidney will result in glomerular or tubular cell injury. It would also play an important role in the pathogenesis in the kidney as well as in the liver. Indeed, ELD in the renal tubules, associated with diabetes, has been shown to be related with renal tubular injury [11]. We coined the term ‘steatonephropathy’ to describe such kidney injury associated with ELD.

The endoplasmic reticulum (ER) is the intracellular organelle of the secretory pathway. The ER plays a central role in regulating protein folding; thus disturbance of ER homeostatic function by toxic reagents or nutrient excess causes the accumulation of misfolded or unfolded protein within the cell [12]. Toxic lipid deposition in the hepatocyte induces ER stress leading to the development or progression of NASH. We have previously reported that ER stress regulates cascade apoptosis of hepatocytes in NASH model mice [13]. However, the association between ELD or ER stress in the kidney and the progression of CKD has not yet been elucidated.

Sodium-glucose cotransporter-2 (SGLT2) inhibitor is a newly introduced type of drug for the treatment of type-2 diabetes. SGLT2 inhibitors block sodium and glucose reabsorption in the renal proximal tubule, leading to glycosuria. In addition to this effect, SGLT2 inhibitors improve glomerular hyperfiltration and insulin resistance [14]. We have also reported that ipragliflozin, an SGLT2 inhibitor, reduced ectopic fat accumulation in the liver and prevented liver fibrosis in NASH model mice [15]. This result indicates that this SGLT2 inhibitor has an additional effect besides glucose metabolism.

We hypothesized that a SGLT2 inhibitor would be effective in improving lipid metabolism in the kidney and would then exert a renoprotective effect by reducing ER stress. The aim of the study was to investigate the effect of ipragliflozin on kidney injury and the involvement of ELD and ER stress in the pathogenesis of ‘steatonephropathy’.

## 2. Results

### 2.1. Effects of Ipragliflozin on Fatty Liver Shionogi ob/ob (FLS-ob/ob) Mice

Metabolic parameters were compared between the control and the ipragliflozin group (Table 1). We observed no difference in the body weight and kidney weight between groups after 12 weeks of treatment. Ipragliflozin improved lipid profile; serum triglyceride was significantly decreased in the ipragliflozin group. Urinary protein excretion adjusted by urinary creatinine showed no significant difference between the groups.

### 2.2. Impact of Ipragliflozin on the Histological Changes in FLS-ob/ob Mice

The effects of ipragliflozin on histological changes including lipid accumulation in renal tubules, glomerular hypertrophy, and interstitial fibrosis were evaluated. Control mice showed lipid droplets in their renal tubules. Treatment with ipragliflozin prevented lipid deposition in the renal tubules (Figure 1). Since SGLT2 inhibitors have been shown to reduce intra-glomerular pressure leading to an improvement of glomerular hypertension [16], we quantified the difference in glomerular size in the control and ipragliflozin groups. The glomerular size was significantly smaller in the ipragliflozin group, indicating an improvement in glomerular hyperfiltration (Figure 2). Masson-trichrome staining of the kidney tissues revealed a significant decrease in the area of interstitial fibrosis in the ipragliflozin group (Figure 3).

### 2.3. Advanced Glycation End Product in Plasma and Kidney

Since ipragliflozin possibly modified the protein glycation resulting in the reduction of ELD, we quantified the levels of advanced glycation end product (AGE) in plasma and kidney homogenates. However, we found no significant differences in the levels of AGE between the control and ipragliflozin group (Appendix A)

### 2.4. Reduced Endoplasmic Reticulum Stress in FLS-ob/ob Mice Treated with Ipraglifilozin

Next, we investigated whether the reduction in lipid droplets affects ER stress in the kidney. The gene expression level of glucose-regulated protein-78 (GRP78), a master regulator of ER stress response, significantly decreased in mice treated with ipragliflozin. The protein expression of GRP78 also decreased in the ipragliflozin group (Figure 4 and Figure 5). Although the mRNA levels of the downstream cascade of GRP78, including protein kinase RNA-activated-like ER kinase (PERK), eukaryotic initiation factor 2α (eIF2α), inositol requiring enzyme 1α (IRE1α), and activating transcription factor 6 (ATF6), did not show significant difference between groups (Figure 4), the other ER stress-associated proteins, including C/EBP homologous protein (CHOP), expression was significantly lower in mice treated with ipragliflozin (Figure 5).

We further explored possible mechanisms underlying the protective effect of ipragliflozin on the induction of ER stress. The mRNA and protein levels of nuclear factor kappa B (NFkB), which plays an important role in the induction of ER stress, did not change after the treatment (Figure 4 and Figure 6). Since we observed a decrease in lipid deposition, we also investigated several proteins involved in lipid accumulation and droplet formation. The expression of diacylglycerol O-acyltransferase (DGAT) 1 was significantly increased in mice treated with ipragliflozin (Figure 6); however, the expression of adipose triglyceride lipase (ATGL) did not change after the treatment.

### 2.5. Ipragliflozin Reduced Cellular Apoptosis in FLS-ob/ob Mice

We then investigated the impact of reduced ER stress on apoptosis in the kidney. The degree of cellular apoptosis was quantified by terminal deoxynucleotidyl transferase dUTP nick end labeling (TUNEL) staining. The ratio of TUNEL-positive cells to all cells was significantly reduced in the ipragliflozin group (Figure 7), suggesting the suppressed apoptotic signal.

## 3. Discussion

In the present study, we demonstrated that ipragliflozin, an SGLT2 inhibitor, prevented ELD in renal tubules and interstitial fibrosis in FLS-*ob/ob* mice. Ipragliflozin also ameliorated ER stress and apoptosis in the kidneys. These data suggested that the improvement of lipid metabolism reduces renal tubular cell apoptosis through regulating ER stress.

SGLT2 inhibitors have been shown to have pleiotropic effects [14,17]. The major function of SGLT2 inhibitors is reducing blood glucose levels by inhibiting the reabsorption of glucose in the proximal tubule, while SGLT2 inhibitors also improve fat metabolism by increasing fat oxidation and ketogenesis. In the present study, we used FLS-*ob/ob* male mice. This strain shows hyperphagia, hypertriglyceridemia, and hyperlipidemia. Although the phenotype in female mice has not been well documented, we have previously reported that male FLS-*ob/ob* mice had severe liver steatosis [13,18,19]. Therefore, we used the same model to investigate the relationship between ectopic lipids in the kidney and kidney injury. We observed ELD in renal tubules as well, which was prevented by ipragliflozin. This finding is in line with our previous report, which revealed less lipid deposition in hepatocytes in NASH model mice, treated with ipragliflozin [15]. In the clinical setting, it has been shown that inhibition of SGLT2 led to the reduction in epicardial fat accumulation, following the reduction in body weight and the improvement in lipid profile and glycemic control [20]. Taking our findings into account, SGLT2 inhibitor seems to be beneficial in reducing ELD in various organs.

Interstitial fibrosis in the FLS-*ob/ob* mice was also prevented by ipragliflozin. ELD induces inflammatory signaling, such as tumor necrosis factor-alfa (TNF-α) and tissue inhibitor of metalloproteinases [13] that is closely related to the progression of fibrosis. In vivo experimentation with high fat diet showed the association between ectopic lipid droplets in renal tubules and TNF-α or transforming growth factor beta, a marker of fibrosis. Therefore, we speculate that ipragliflozin attenuated the progression of interstitial fibrosis through the reduction of ELD in renal tubules.

ER stress-associated genes and/or proteins and the apoptotic signal were downregulated in FLS-*ob/ob* mice treated with ipragliflozin. Previous in vitro experiments showed that high glucose levels induced lipid deposition in renal tubular cells and tubular injury. In addition to the expression of TNF-α, lipid metabolism-associated markers such as adipose differentiation-related protein and sterol regulatory element binding protein-1 were upregulated in renal tubular cells treated with high glucose, which was accompanied by lipid deposition [11]. However, to our knowledge, there has been no report showing the association between ELD and induction of ER stress in the kidney. In the present study, we also observed a significant increase in the protein expression of DGAT1 in mice treated with ipragliflozin. DGAT1 is an enzyme that catalyzes the synthesis of triglyceride from fatty acids to protect cells from lipotoxicity. DGAT1 is exclusively localized in the ER and serves to prevent ER stress under exposure to excess fatty acids [21]. Accumulated triglyceride is subsequently hydrolyzed by ATGL, hormone sensitive lipase, or monoglyceride lipase [22]. Since there were no significant differences in the expression of ATGL, lipolysis by the treatment of ipragliflozin might be dependent on other pathways. We also measured the levels of AGE in plasma or kidney tissue to investigate the possible association of the protein glycation and the induction of ER stress. However, we could not find any differences in the levels of AGE between the groups. In addition, the mRNA and protein expressions of NFkB, one of the major transcription factors downstream of AGE, were not changed by ipragliflozin.

ER stress response is mediated by three different sensors, namely PERK, IRE1α, and ATF6 [23]. GRP78 is a master regulator of the ER stress response and acts by activating these three sensors, and both eIF2α and CHOP are downstream mediators of PERK that control cell death. We observed that mice, treated with ipragliflozin, showed lower expression of GRP78 and CHOP, accompanied by a reduction in apoptotic cells, without a significant change in IRE1α or ATF6. These results indicate that ipragliflozin suppresses the apoptotic signal through the GRP78 and CHOP pathway. In a previous investigation in the liver, ectopic lipid in the hepatocyte induced cellular apoptosis through induction of ER stress, followed by an upregulation of CHOP [12], which resulted in the progression of fibrosis [13]. ER stress, induced by ELD, is considered to be the common pathogenic mechanism underlying both steatohepatitis and steatonephropathy. Since ipragliflozin reduced ELD in the renal tubular cells, we speculated that ipragliflozin prevented kidney fibrosis and cell apoptosis by improving lipid profiles with attenuated ER stress.

In this study, glomerular hypertrophy was improved in the ipragliflozin group. Glomerular hypertrophy, a symptom often seen in diabetic nephropathy or in obese patients, reflects glomerular hyperfiltration, which is strongly associated with proteinuria or albuminuria. Microalbuminuria, defined by urinary albumin-to-creatinine ratio, is a well-known early marker for the progression of kidney diseases [24]. Proteinuria and albuminuria induce proximal tubular injury, leading to interstitial fibrosis, and are seen as independent risk factors for reducing renal function. The other markers that indicate the early progression of kidney diseases are neutrophil gelatinase associated lipocalin and angiopoietin-2 [25,26,27]. Although we could not measure urinary albumin levels and these early indicators, both groups showed overt proteinuria suggesting kidney injury in our model had already progressed to some degree. Since urinary parameters showed no significant difference between groups, we speculated that ipragliflozin suppressed further apoptosis or interstitial fibrosis through improving lipid profile.

There are some limitations in this study. SLGT2 reabsorbs sodium in the proximal tubule as well. Sodium absorption/excretion might affect blood pressure regulation. That would be another aspect related to CKD. SGLT2 inhibition modifies glycemic profiles, thus improved glycemic profiles by ipragliflozin are possibly affected ER stress, apoptosis, and interstitial fibrosis in this study. ELD has been shown to be associated with insulin resistance and cardiovascular diseases [28]. Although similar conditions might exist in renal tubules in the present study, we could not investigate the vascular factor in the etiology of kidney damage.

In conclusion, we demonstrated that ER stress is involved in the pathogenesis of steatonephropathy, and that an SGLT2 inhibitor, ipragliflozin, attenuates the progression of kidney injury by modifying the lipid profile. Modification of lipid profiles could be a new therapeutic approach to prevent the progression of CKD in patients with NASH.

## 4. Materials and Methods

### 4.1. Animals and Experimental Designs

Male FLS-*ob/ob* mice were obtained from Shionogi Research Laboratories (Shionogi Research Laboratories, Shiga, Japan). This strain is well characterized and established as the most similar model to human NASH [16]. The mice were housed in a room maintained at a controlled temperature of 24 ± 2 °C under a 12-h light-dark cycle with ad libitum water and standard pellet chow (CE-2, 4.6% fat; CLEA Japan, Tokyo, Japan). All animal experiments were performed in accordance with the Animal Experimentation Guidelines of Tottori University (h26-Y-008, 8 May 2014).

Twelve FLS-*ob/ob* mice at 12-week old were divided into two groups: the control group (*n* = 6) and ipragliflozin group (*n* = 6). The mice were administered 1 mg/kg Ipragliflozin or vehicle via gavage every day for 12 weeks. Ipragliflozin at a dose of 1 mg/kg is often used in clinical settings, and has no toxic side effects to mice at this dose [13]. After 12 weeks, for metabolic studies to collect urine, mice were individually housed in metabolic cages with ad libitum water and standard pellet chow. Twenty four-hour urine was collected and stored at −80 °C. After metabolic studies, the mice were sacrificed under pentobarbital anesthesia injection (Dainippon Sumitomo Pharma, Osaka, Japan). The blood was obtained from the right ventricle, subsequently centrifuged for 20 min at 2000× *g*, and the plasma samples were stored at −80 °C. Harvested kidneys were snap-frozen in liquid nitrogen and kept at −80 °C. For histological analysis, the kidneys were fixed in 10% buffered formalin (Wako Pure Chemical Industries, Osaka, Japan) and embedded in paraffin (Wako Pure Chemical Industries, Osaka, Japan).

### 4.2. Histological Analysis

Four-μm-thick sections of the kidneys were prepared for histological analyses. The sections were stained with periodic acid-Schiff (PAS) for the evaluation of glomerular size and lipid droplets in renal tubules, and Masson-trichrome for the evaluation of interstitial fibrosis. Lipid droplets were quantified as the relative area of droplets to the tissue area. At least three fields from each mouse (magnification ×40) were randomly quantified. The glomerular size was quantified as the area of Bowman’s capsule. A number of 30 glomeruli in each mouse were randomly quantified, and the largest 20 glomeruli were used for the analyses. The interstitial fibrosis area was quantified in three randomly selected fields from each mouse (magnification ×20) and the percentage of fibrotic area was measured. ImageJ software (U.S. National Institute of Health, Bethesda, MD, USA) was used for the quantification.

### 4.3. Quantification of AGE in the Plasma and Kidney

The levels of AGE in plasma and kidney homogenates were measured by ELISA (OxiSelect^TM^ Advanced Glycation End Product Competitive ELISA Kit, Cell Biolabs, Inc., San Diego, CA, USA) according to the manufacturer’s protocol. Plasma and 100 μg protein of kidney homogenates were diluted in phosphate-buffered saline and used for the quantification.

### 4.4. Apoptosis Assay in Tissues

Apoptotic cells in kidney tissues were quantified with TUNEL assay (In Situ Cell Death Detection Kit; Roche, Basel, Switzerland). Four-μm-thick sections were labeled according to the manufacturer’s protocol. The sections were mounted with ProLong Antifade (Invitrogen–Molecular Probes, Carlsbad, CA, USA) containing 4′,6-diamidino-2-phenylindole (DAPI). Apoptotic cells were quantified as the percentage of the number of TUNEL positive cells to the all cells. At least three fields were randomly selected from each mouse (magnification ×40). ImageJ software (U.S. National Institute of Health, Bethesda, MD, USA) was used for the quantification.

### 4.5. Gene Expression Analysis

Tissue samples were homogenized and total RNA was extracted with the RNeasy Mini Kit (Qiagen, Hilden, Germany) according to the manufacturer’s protocol. RNA concentration was determined by nano-drop-1000 spectrophotometer (Thermo Fisher Scientific Inc., Tokyo, Japan), and total RNA of 2 μg was used to perform reverse transcriptase reaction in a final volume of 20μL containing 1× RT buffer, 4 mM dNTP mix, 1× RT random primer, 50 units multiscribe reverse transcriptase, 20 units RNase inhibitor, and nuclease-free water at 25 °C for 10 min, followed by 37 °C for 120 min and 85 °C for 5 min. Changes in target genes mRNA levels were determined by quantitative real-time polymerase chain reaction (PCR) in 20 μL aliquots containing 1 μL reverse transcriptase reaction products, 4 μL LightCycler FastStart DNA MasterPLUS SYBER-Green I (Roche Diagnostics, Tokyo, Japan), 0.5 μM each primer (shown in Table 2) and 14.6 μL nuclease free water and run on the real-time PCR Lightcycler 1.5 complete system (Roche Diagnostics, Tokyo, Japan). Thermal cycling was initiated with a first denaturation step at 95 °C for 10 min followed by 45 cycles of 95 °C for 10 s, 60 °C for 10 s, and 72 °C for 10 s. The cycle passing threshold (Ct) was recorded for mRNA by LightCycler software version 3.5.28 (Roche Diagnostics, Tokyo, Japan). Beta-actin was used as the internal control for data normalization.

### 4.6. Western Blot Analysis

Proteins were extracted from grinded kidney tissues in ice-cold RIPA buffer (Thermo Fisher Scientific, Tokyo, Japan), containing protease and phosphatase inhibitor (Roche Diagnostics, Tokyo, Japan). Protein concentrations were measured with Pierce 660 nm Protein Assay Reagent (Thermo Fisher Scientific, Tokyo, Japan). Samples were mixed with the Laemmli buffer and 2-ME followed by boiling at 95 °C for 5 min. Forty μg of protein were loaded for sodium dodecylsulfate-polyacrylamide gel electrophoresis and transferred to nitrocellulose membrane. After blocking with 5% skim milk in tris-buffered saline, the membrane was incubated with rabbit anti-GRP78 (1:1000; abcam), mouse anti-CHOP (1:1000; CST), rabbit anti-NFkB (1:1000; GeneTex), rabbit anti-DGAT1 (1:500; abcam), rabbit anti-ATGL (1:1000; CST), or rabbit anti-β-actin (1:2000; CST) overnight at 4 °C. The membrane was washed with tris-buffered saline and incubated with their corresponding horseradhish peroxidase-conjugated secondary antibodies (1:5000) for 1 h at room temperature. Signals were visualized using Clarity Western ECL Substrate (Bio-Rad Laboratories, Tokyo, Japan) and image analyzer (LAS-3000 mini; Fujifilm, Tokyo, Japan). The protein expression was normalized with the signal obtained with β-actin expression.

### 4.7. Statistical Analysis

The unpaired *t*-test was performed to assess the differences between the groups. *p*-values less than 0.05 were considered as significant. GraphPad Prism (7.0. for Windows, GraphPad Software, San Diego, CA, USA) was used for the statistical analysis. All values are expressed as the mean ± SEM.

## Figures and Tables

**Figure 1 ijms-21-00190-f001:**
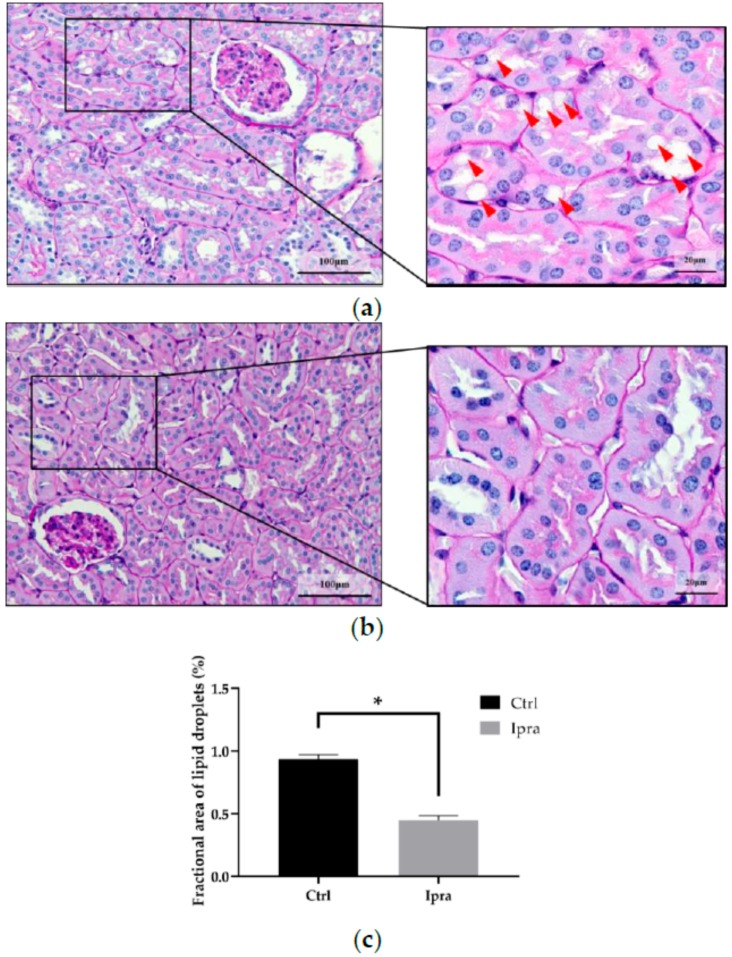
Lipid deposition in renal tubule with or without ipragliflozin. Representative images of Periodic acid-Schiff staining on kidneys paraffin-embedded 4-μm-sections from FLS-*ob/ob* mouse treated with (**a**) vehicle or (**b**) ipragliflozin 1 mg/kg. Magnified images from both groups were also shown. Lipid droplets could be observed in renal tubular epithelial cells (arrowheads) from the control mouse kidneys, in contrast with the sparse lipid droplets in the mice treated with ipragliflozin. These results indicated the effect of ipragliflozin on reducing lipid deposition in the renal tubules. (**c**) Quantification of the amount of lipid droplets. Fractional area of lipid droplets was calculated as the ratio of the total amount of lipid droplets area to the whole tissue area. The quantification is based on randomly captured three fields from six different mice in each group. Bars indicate average ± SEM. * *p* < 0.05 (unpaired *t*-test). Ctrl, control group; Ipra, ipragliflozin group.

**Figure 2 ijms-21-00190-f002:**
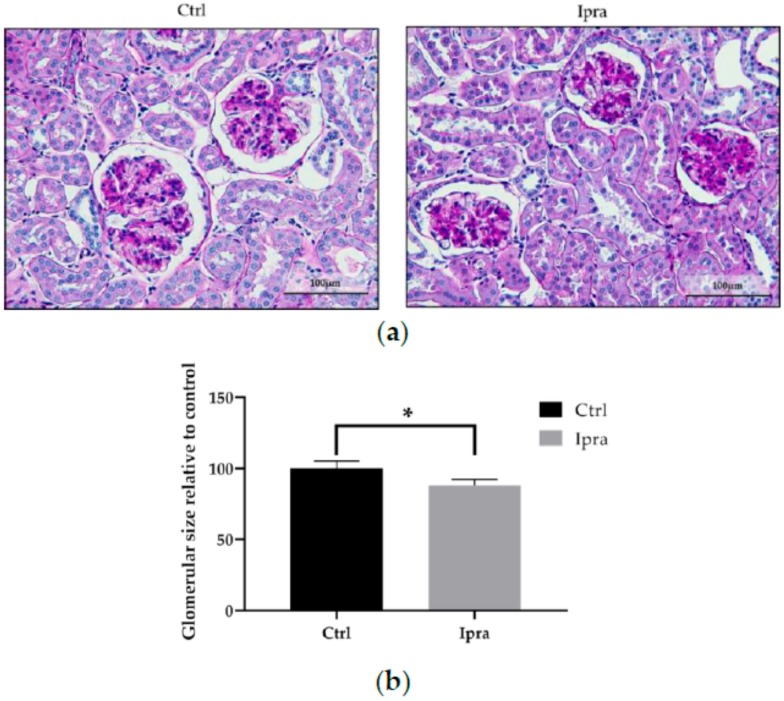
Effect of ipragliflozin on glomerular size. (**a**) Representative images of Periodic acid-Schiff staining on FLS-*ob/ob* mouse kidney paraffin-embedded sections. (**b**) Quantification of glomerular size. The results were expressed as the Bowman’s capsule area relative to control group. The quantification is based on at least 20 glomeruli from 6 different mice in each group. Bars indicate average ± SEM. * *p* < 0.05 (unpaired *t*-test). Ctrl, control group; Ipra, ipragliflozin group.

**Figure 3 ijms-21-00190-f003:**
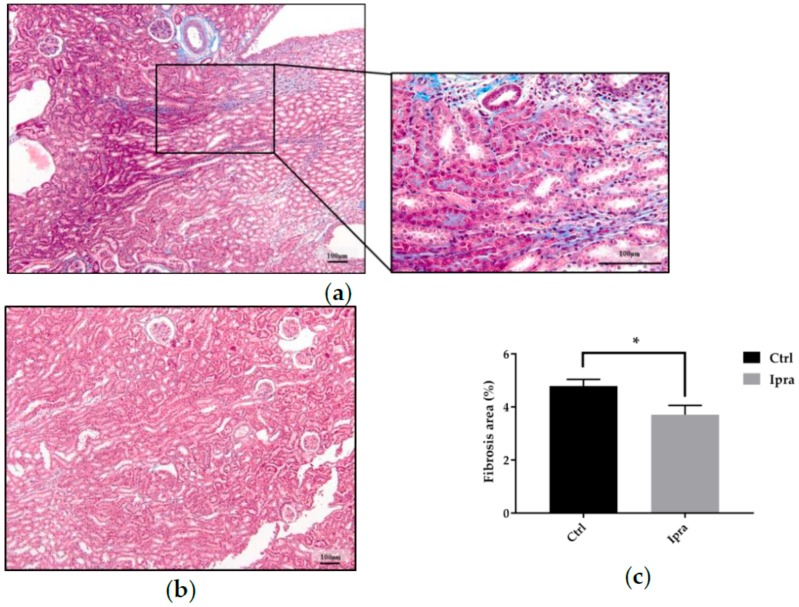
Effect of ipragliflozin on interstitial fibrosis. (**a**) Representative images of Masson-trichrome staining on kidney paraffin-embedded sections from FLS-*ob/ob* mouse in the control group with a high magnification image. (**b**) Representative image of Masson-trichrome staining on kidney paraffin-embedded sections from FLS-*ob/ob* mouse in the ipragliflozin group. (**c**) Quantification of the area of fibrosis. The results were expressed as the percentage of fibrotic area to the whole area. The quantification is based on randomly captured three fields from six different mice in each group. Bars indicate average ± SEM. * *p* < 0.05 (unpaired *t*-test). Ctrl, control group; Ipra, ipragliflozin group.

**Figure 4 ijms-21-00190-f004:**
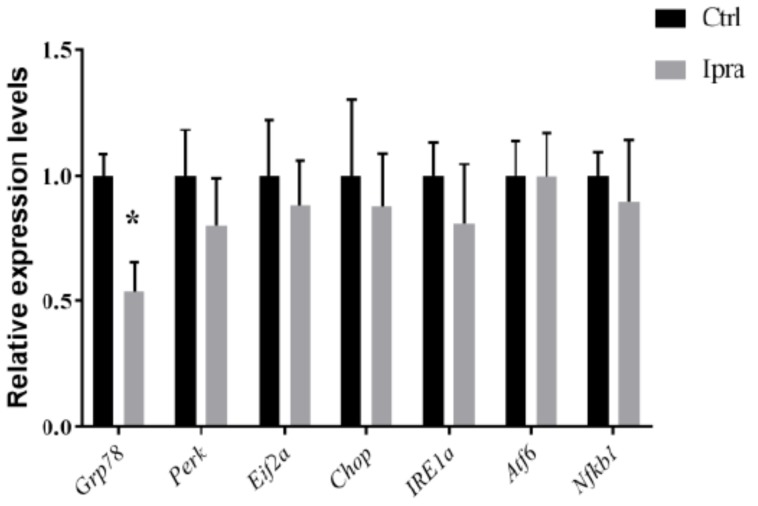
Endoplasmic reticulum stress-associated gene expressions. Relative gene expression levels of Grp78, Perk, Eif2a, Chop, Ire1a, Atf6, and Nfkb1 as quantified by quantitative reverse transcription-polymerase chain reaction (RT-qPCR) in kidney tissues from FLS-*ob/ob* mouse. Beta-actin was used as an internal control. Levels are expressed relative to control group. Bars indicate average ± SEM. * *p* < 0.05 (unpaired *t*-test). Ctrl, control group; Ipra, ipragliflozin group.

**Figure 5 ijms-21-00190-f005:**
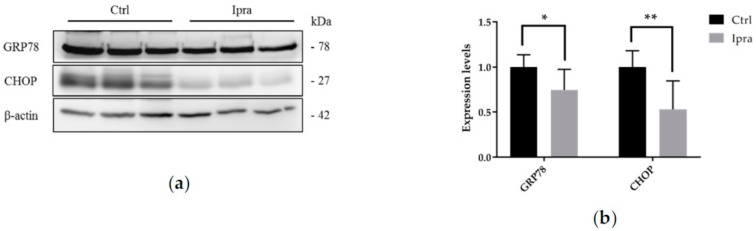
Endoplasmic reticulum stress-associated protein expression. (**a**) Western blot analysis for endoplasmic reticulum-associated proteins on kidney tissues from FLS-*ob/ob* mouse. Beta-actin is used as a loading control. (**b**) Quantification of western blot signal intensities, expressed relative to control group. Bars indicate average ± SEM. * *p* < 0.05; ** *p* < 0.01 (unpaired *t*-test). Ctrl, control group; Ipra, ipragliflozin group.

**Figure 6 ijms-21-00190-f006:**
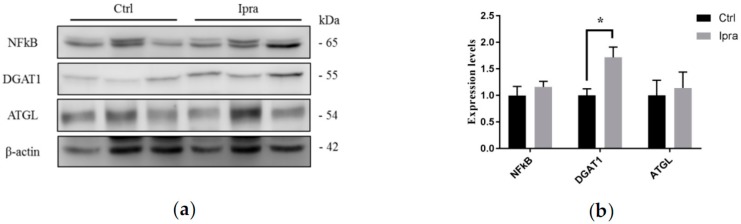
Expressions of lipid-associated protein. (**a**) Western blot analysis for NFkB, DGAT1, and ATGL on kidney tissues from FLS-*ob/ob* mouse. Beta-actin is used as a loading control. (**b**) Quantification of western blot signal intensities, expressed as relative level to control group. Bars indicate average ± SEM. * *p* < 0.05; ** *p* < 0.01 (unpaired *t*-test). Ctrl, control group; Ipra, ipragliflozin group.

**Figure 7 ijms-21-00190-f007:**
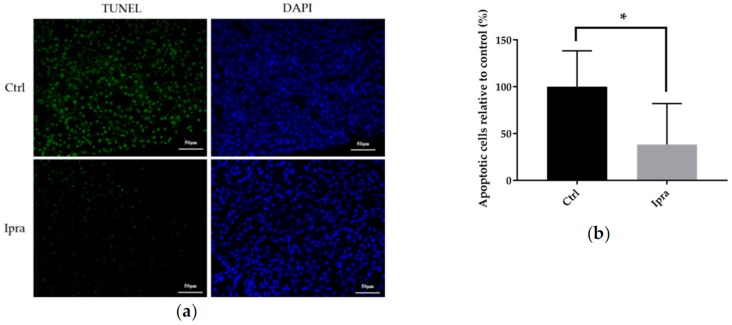
Effect of ipragliflozin on renal cellular apoptosis in FLS-*ob/ob* mice. (**a**) Representative images of TUNEL staining on FLS-*ob/ob* mouse kidney paraffin-embedded sections. (**b**) Quantification of apoptotic cells. The percentage of TUNEL positive cells to DAPI positive cells was expressed as relative level to control group. The quantification is based on randomly captured three fields from six different mice in each group. Bars indicate average ± SEM. * *p* < 0.05 (unpaired *t*-test). TUNEL, terminal deoxynucleotidyl transferase dUTP nick end labeling; DAPI, 4′,6-diamidino-2-phenylindole; Ctrl, control group; Ipra, ipragliflozin group.

**Table 1 ijms-21-00190-t001:** Metabolic characteristics in the two mice groups

Parameter	Control	Ipragliflozin	*p*-Value
BW (g)	51.6 ± 2.6	53.4 ± 1.6	0.573
Kidney weight/BW	1.3 ± 0.2	1.3 ± 0.1	0.816
Urine volume (mL/day)	7.6 ± 1.9	3.3 ± 1.2	0.086
Blood glucose (mg/dL)	295.7 ± 64.4	183.3 ± 32.2	0.150
Serum triglyceride (mg/dL)	255.6 ± 39.2	154.0 ± 14.0	0.041 *
Creatinine clearance (mL/min/kg)	7.3 ± 1.6	5.7 ± 2.2	0.568
Urinary protein (g/g/Cr)	13.2 ± 1.3	10.2 ± 2.2	0.256

Abbreviations. BW, body weight. * *p* < 0.05 comparison between two groups.

**Table 2 ijms-21-00190-t002:** List of primers for qPCR analysis

Gene Product	Forward Primer (5′-3′)	Reverse Primer (5′-3′)
Grp78	GAG GCG TAT TTG GGA AAG AAG G	GCT GCT GTA GGC TCA TTG ATG
Perk	TAT GCT ACG CAC ACG GGA CAA	ACT CGT TCC ATC TGG GTG CTG
Eif2a	GCA CTA CAT TGC AAC AAA TGG	CAA ACA CAG GAT CAC ACT TCA G
Chop	AGC TGG AAG CCT GGT ATG AG	AGC TAG GGA TGC AGG GTC AA
Ire1a	AAG GCT GGA TGG CAC CAG AG	ATG TTG GCC TGT CGG TAG AGA
Atf6	TCG CCT TTT AGT CCG GTT CTT	GGC TCC ATA GGT CTG ACT CC
Nfkb1	ATTCCGCTATGTGTGTGAAGG	GTGACCAACTGAACGATAACC
Actb	GCA TCC TCA CCC TGA AGT A	TGT GGT GCC AGA TTT TCT CC

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
