# Peer review of "Ipragliflozin Ameliorates Endoplasmic Reticulum Stress and Apoptosis through Preventing Ectopic Lipid Deposition in Renal Tubules"

_ijms, 2019, doi:10.3390/ijms21010190_

Round 1
Reviewer 1 Report
I have no competing interests.
It is an interesting and relevant article. I consider it a useful contribution in its field.
The figure on Page 3 on kidney biopsy needs footnote and figure legend, need better clarifications on the staining and description.
What is P value for figure 4.
Discussion section needs more additional sentences on future implications.
Some revision of the English language is needed.
Author Response
Reply to reviewer 1
It is an interesting and relevant article. I consider it a useful contribution in its field.
The figure on Page 3 on kidney biopsy needs footnote and figure legend, need better clarifications on the staining and description.
Author’s reply: It is a pleasure to receive such comment from the reviewer. We have added some explanations to the figure legend. The bar graph was changed to absolute value for better understanding. The figure legend was changed to “Figure 1. Lipid deposition in renal tubule with or without Ipragliflozin. Representative images of Periodic acid-Schiff staining on kidneys paraffin-embedded 4-μm-sections from FLS-ob/ob mouse treated with (a) vehicle or (b) Ipragliflozin 1mg/kg. Magnified images from both groups were also shown. Lipid droplets could be observed in renal tubulares epithelial cells (arrowheads) from control mice kidneys, in contrast to the sparse lipid droplets observed in control mice were absent in mice treated with Ipragliflozin. These results indicated the effect of Ipragliflozin on reducing the lipid deposition in renal tubules. (c) Quantification of the amount of lipid droplets. Fractional area of lipid droplets was calculated as the ratio of the total amount of lipid droplets area to the whole tissue area. The quantification is based on randomly captured 3 fields from 6 different mice in each group. Bars indicate average ± SEM. *P<0.05 (Unpaired t test). Ctrl, control group; Ipra, Ipragliflozin group.”
What is P value for figure 4.
Author’s reply: The P value of less than 0.05 was considered as significant. No significant differences were found in AGE levels between groups. We have added “No significant differences were found between each group.” in the figure legend.
Discussion section needs more additional sentences on future implications.
Author’s reply: Thank you for the comment. We have added “Modification of lipid profiles could be a new therapeutic approach to prevent the progression of CKD in patients with NASH.” in the conclusion.
Some revision of the English language is needed.
Author’s reply: I apologize for the poor English. The manuscript underwent English check. Some grammatical changes were made according to the check.

Reviewer 2 Report
It is a well chosen subject of this article, important clinically. Congratulations to the authors. After reading this article submitted to me for review, however, it occurred to me observations and comments. The described comments and suggested changes in the text lead to a better understanding of the theme and will increase readers' interest in this topic. Here they are.
Main points:
In the “Introduction” section the authors wrote: “Metabolic syndrome including diabetes, hypertension, obesity, and dyslipidemia is strongly associated with the development and progression of CKD, and diabetic nephropathy is a well-known cause of end-stage renal disease”. In his section complete lack of information on one of the most important etiological factors for the development of diabetes mellitus - pancreatic injury and causes of glucose metabolism disorders. To better understand the topic should be extended introduction about the latest hormone physiology (for example PMID: 25716961 and many more). It is also important to discuss. This should be discussed. Such changes will result in the introduction of a better understanding of this topic. It is also necessary to add information about studies on new indicators monitoring kidney function (for example Angiopoietin-2 (PMID: 27022209), urine NGAL PMID: 27513835, PMID: 28050059, PMID: 28613246). Authors presents in Table 1. metabolic characteristics - among others creatinine clearance and urinary protein. Authors do not present the albumin/creatinine ratio in discussed research. This problem needs to be discussed (PMID: 29324682). The authors should refer to research on this. Authors should discuss the problem to refer to research regarding - selected laboratory markers of glomerular and tubular damage in T2DM patients with early stages of chronic kidney disease (G1/G2, A1/A2) for their associations with A2 albuminuria and early decline in the estimated glomerular filtration rate (eGFR) (PMID: 30158836). The discussion is too short. Over 50% of end-stage renal disease (ESRD) patients die of cardiovascular disease. ESRD patients treated with maintenance hemodialysis are repeatedly exposed to oxidative stress. (PMID: 31583040). Based on the findings, there are weak associations between nutritional status and selected redox parameters in hemodialyzed patients. Further studies are needed to establish if diet modifications and adequate nutritional status can positively impact the antioxidant capacity in this group of patients, especially in the context of non-alcoholic steatohepatitis (NASH) described by the authors. Malnutrition-inflammation-atherosclerosis syndrome is one of the causes of increased mortality in chronic kidney disease (CKD). The aim of the study was to assess the inflammation and nutritional status of patients in end-stage kidney disease treated with maintenance hemodialysis. The strict dietary restrictions in CKD, among in the context of non-alcoholic steatohepatitis (NASH) are difficult to fulfill, and if strictly followed, may lead to protein-calorie malnutrition (PMID: 29324682). The authors should refer to research on this. It has been shown that ischemic preconditioning of distant organs such as the kidney, intestine or limb may protect the heart as effectively as cardiac preconditioning itself. Direct ischemic preconditioning attenuated the severity of for example acute pancreatitis (PMID: 18622049). It was found a diminution of histological signs of pancreatic damage, as well as, an improvement of pancreatic blood flow and DNA synthesis. In contrast to direct ischemic preconditioning, remote ischemic preconditioning is without effect for example on pancreatic exocrine secretion and does not reduce the severity of ischemia/reperfusion-induced pancreatitis. It can be similar with the liver in the context of NASH. The authors should mention this, especially since they did not discuss the vascular factor in the etiology of organ damage.This extension and explanation of the topic will clarify the path that led researchers to the presented conclusions.
Author Response
Reply to reviewer 2
It is a well chosen subject of this article, important clinically. Congratulations to the authors. After reading this article submitted to me for review, however, it occurred to me observations and comments. The described comments and suggested changes in the text lead to a better understanding of the theme and will increase readers' interest in this topic. Here they are.
Author’s reply: It is a pleasure to receive the comments together with the constructive advice to improve the manuscript. Our reply to each comment is below.
Main points:
In the “Introduction” section the authors wrote: “Metabolic syndrome including diabetes, hypertension, obesity, and dyslipidemia is strongly associated with the development and progression of CKD, and diabetic nephropathy is a well-known cause of end-stage renal disease”. In his section complete lack of information on one of the most important etiological factors for the development of diabetes mellitus - pancreatic injury and causes of glucose metabolism disorders. To better understand the topic should be extended introduction about the latest hormone physiology (for example PMID: 25716961 and many more). It is also important to discuss. This should be discussed. Such changes will result in the introduction of a better understanding of this topic.
Author’s reply: Thank you for the suggestions. We have extended the section introducing the peptidyl hormones involved in the development of diabetes. We have added “Several peptidyl hormones mainly produced in gastrointestinal tract are reported to be associated with the development of CKD in the context of diabetic nephropathy [ref]. Glucagon-like peptide-1 and its related peptide, dipeptidyl peptidase IV, are the major therapeutic targets for diabetes and diabetic nephropathy [ref]. In contrast to the current understanding and progression in therapeutic approach to diabetic nephropathy, mechanisms of CKD caused by dyslipidemia remain to be clarified.”
It is also necessary to add information about studies on new indicators monitoring kidney function (for example Angiopoietin-2 (PMID: 27022209), urine NGAL PMID: 27513835, PMID: 28050059, PMID: 28613246). Authors presents in Table 1. metabolic characteristics - among others creatinine clearance and urinary protein. Authors do not present the albumin/creatinine ratio in discussed research. This problem needs to be discussed (PMID: 29324682). The authors should refer to research on this. Authors should discuss the problem to refer to research regarding - selected laboratory markers of glomerular and tubular damage in T2DM patients with early stages of chronic kidney disease (G1/G2, A1/A2) for their associations with A2 albuminuria and early decline in the estimated glomerular filtration rate (eGFR) (PMID: 30158836).
Author’s reply: I agree that urinary albumin is an early indicator of kidney injury in diabetic nephropathy and that there are new markers for predicting kidney injury such as NGAL and angiopoietin-2. We incorporated these points and added some sentences and references as the reviewer suggested. We have added “Microalbuminuria, defined by urinary albumin to creatinine ratio, is a well-known early marker for the progression of kidney disease [ref]” and “The other markers that indicate the early progression of kidney disease are NGAL and angiopoietin-2 [ref]. Although we could not measure urinary albumin levels and these early indicators, both the groups showed overt proteinuria suggesting kidney injury in our model was already progressed to some degree.” in the discussion section.
The discussion is too short. Over 50% of end-stage renal disease (ESRD) patients die of cardiovascular disease. ESRD patients treated with maintenance hemodialysis are repeatedly exposed to oxidative stress. (PMID: 31583040). Based on the findings, there are weak associations between nutritional status and selected redox parameters in hemodialyzed patients. Further studies are needed to establish if diet modifications and adequate nutritional status can positively impact the antioxidant capacity in this group of patients, especially in the context of non-alcoholic steatohepatitis (NASH) described by the authors. Malnutrition-inflammation-atherosclerosis syndrome is one of the causes of increased mortality in chronic kidney disease (CKD). The aim of the study was to assess the inflammation and nutritional status of patients in end-stage kidney disease treated with maintenance hemodialysis. The strict dietary restrictions in CKD, among in the context of non-alcoholic steatohepatitis (NASH) are difficult to fulfill, and if strictly followed, may lead to protein-calorie malnutrition (PMID: 29324682). The authors should refer to research on this.
Author’s reply: I agree that ESRD patients with maintenance hemodialysis are exposed to oxidative stress and inflammation, and the contribution of nutritional status to redox parameters may relatively weak in such patients. However it does not seem to be directly applied to the mechanism of lipid deposition-induced ER stress. The aim of the study was not to assess the inflammation and nutritional status in ESRD patients, but to investigate the association of ectopic lipid to ER stress and therapeutic effect of Ipragliflozin. Malnutrition or dietary restrictions are not the main topic discussed in the study. Although nutritional approach in CKD patients is an important to issue to be solved, it is beyond the scope of this study.
It has been shown that ischemic preconditioning of distant organs such as the kidney, intestine or limb may protect the heart as effectively as cardiac preconditioning itself. Direct ischemic preconditioning attenuated the severity of for example acute pancreatitis (PMID: 18622049). It was found a diminution of histological signs of pancreatic damage, as well as, an improvement of pancreatic blood flow and DNA synthesis. In contrast to direct ischemic preconditioning, remote ischemic preconditioning is without effect for example on pancreatic exocrine secretion and does not reduce the severity of ischemia/reperfusion-induced pancreatitis. It can be similar with the liver in the context of NASH. The authors should mention this, especially since they did not discuss the vascular factor in the etiology of organ damage. This extension and explanation of the topic will clarify the path that led researchers to the presented conclusions.
Author’s reply: I understand the point. Vascular factors in the etiology of organ damage were not investigated in the study. Since ELD has been shown to be associated with insulin resistance and cardiovascular diseases, similar pathophysiological condition might exist in renal tubules. Unfortunately we could not investigate insulin resistance or vascular changes in the present study, this point was considered to be the limitation of our study. We have incorporated this in the limitation.

Round 2
Reviewer 2 Report
The authors responded to the suggestions. The current version is acceptable
Author Response
Reply to reviewer 2
The authors responded to the suggestions. The current version is acceptable
Author’s reply: We are glad to hear that. Thank you very much for reviewing the manuscript.

This manuscript is a resubmission of an earlier submission. The following is a list of the peer review reports and author responses from that submission.
Round 1
Reviewer 1 Report
The manuscript by Hosokawa.K et al., entitled - Ipragliflozin ameliorates endoplasmic reticulum 3 stress and apoptosis through preventing ectopic lipid 4 deposition in renal tubules is an interesting observation. Here the authors have tried to evaluate the role of ectopic lipid deposition and also ER stress in induction of Chronic Kidney Disease (CKD), and have made an attempt to investigate the efficacy of a sodium glucose cotransporter-2 inhibitor, Ipragliflozin. It is an interesting observation and they have tried to make a connection between the use of Ipragliflozin and reduction in ER stress markers like gpr78 and CHOP, the number of apoptotic cells, and reduction in interstitial fibrosis in male FLS-ob/ob mice model.
There are some concerns which needs to be addressed, before this work can be published. Kindly perform the suggested experiments as that will enhance the quality of the manuscript.
The authors have used male FLS-ob/ob mice which are 12 week old, it would be ideal to provide justification atleast why only male animals have been used for this work, or if authors have have data with female mice models for the same duration, using Ipragliflozin would provide a better insight. Also we need to know whether this dose of 1 mg/Kg currently used in the study has not got any systemic toxic effects following 12 weeks of use, mostly in liver and heart, so it would be ideal to measure AST and ALT levels, plasma creatinine, caspase-3 cleavage and fibrosis markers like TGF beta expression in liver and also heart. The mechanism behind Ipragliflozin and reduction in apoptosis, ER stress and interstitial fibrosis is not clear, so authors need to explore more the cellular mechanisms behind Ipragliflozin mediated protection during chronic kidney disease. The authors have measured only blood glucose following Ipragliflozin administration but haven’t checked the levels protein glycation, are these effects the authors observing due to just prevention of glucose uptake or due to reversal of protein glycation modification. It would be ideal to identify markers which give a surrogate mark of the glycation end product formation. Another possibility is that Receptors for AGEs or TLRs could be playing a role in ER stress induction, and one of the major transcription factor that is known to act as a downstream of these upstream regulators is NF kappa B. So it would be nice to measure the mRNA and protein expression of NF Kappa B and also the transcription function. With the administration of Ipragliflozin, authors see a decrease in lipid deposition, and some of the major proteins involved in lipid accumulation and droplet formation is ATGL, DGAT and Perilipin, so it would be ideal to measure their protein expression following treatment.Reviewer 2 Report
The work is well structured, but it is necessary toreconsider the following aspect:
- Plagiarism detected is 36 %, so it should be reduced.
- Methods are not correctly located, must be included after introduction.
- English must be improved.
- I recomend check how to write the numbers, sometimes they are
included with lyrics and sometimes with numbers
- The bibliography must be update and extended significantly
the sections are not correctly arranged. Finally, I advise you to check English.